# The Risk of Psychological Stress on Cancer Recurrence: A Systematic Review

**DOI:** 10.3390/cancers13225816

**Published:** 2021-11-19

**Authors:** Hyeon-Muk Oh, Chang-Gue Son

**Affiliations:** 1College of Korean Medicine, Daejeon University, Daejeon 35235, Korea; oh033@naver.com or; 2Liver and Immunology Research Center, Daejeon Korean Medicine Hospital of Daejeon University, Daejeon 35235, Korea

**Keywords:** psychological stress, life events, cancer recurrence, systematic review

## Abstract

**Simple Summary:**

Cancer imposes the largest clinical and economic burden, with an estimated 19.3 million cancer incidents and almost 10.0 million cancer deaths in 2020 worldwide. Despite advances in early diagnosis before cancer progression, there are still feared concerns about recurrence after primary treatments. Psychological stress has been known to contribute to the development and progression of cancer; however, its effect on cancer recurrence remains inconclusive. To determine the association between psychological stress and the risk of cancer recurrence, this review aims to systematically evaluate the relevant studies. This study provides comprehensive information about the importance of psychological stress on cancer recurrence and provides reference data to clinicians and scientists for further studies.

**Abstract:**

Cancer recurrence is a significant clinical issue in cancer treatment. Psychological stress has been known to contribute to the incidence and progression of cancer; however, its effect on cancer recurrence remains inconclusive. We conducted a systematic review to examine the current evidence from the Medline (PubMed), Embase and Cochrane Library up to May 2021. Among 35 relevant articles, a total of 6 studies (10 data points) were finally selected, which enrolled 26,329 patients (26,219 breast cancer patients except hepatocellular carcinoma patients in 1 study), 4 cohort studies (8 data points) and 2 RCTs (2 data points). Among the 8 data points in cohort studies, four psychological stress-related factors (two ‘anxiety’, one ‘depression’, and one ‘hostility’) were shown to be moderately related with the risk for cancer recurrence, while ‘loss of partner’ resulted in opposite outcomes. The ‘emotional‘ and ‘mental’ health factors showed conflicting results, and an RCT-derived meta-analysis proved the positive efficiency of psychotherapies in reducing the cancer recurrence risk among breast cancer patients (HR = 0.52; 95% CI 0.33–0.84). Despite the limitations, this study produces comprehensive information about the effect of psychological stress on cancer recurrence and provides reference data to clinicians and scientists for further studies.

## 1. Introduction

Cancer imposes the largest clinical and economic burden and is the leading cause of death, leading to an estimated 19.3 million cancer incidents and almost 10.0 million cancer deaths in 2020 worldwide [1]. Among the whole process of tumor development and its clinical treatments/management, the metastasis and recurrence of cancer are the most significant clinical issues that determine the final outcome of cancer patients [2]. These two malicious episodes are closely connected and are responsible for 66.7% of cancer-related deaths, especially among patients with solid tumors [3].

Cutting-edge and predictive technologies can now be used to make an early diagnosis of cancer prior to the advanced stage [4]; however, there are still concerns about recurrence after the primary treatment of cancers [5]. In general, the 5-year recurrence rates are reported to be 7, 11, and 13% after standard treatment for patients with stage I, II, and III breast cancer, respectively [6], and 25–40% after surgery for colorectal cancer [7]. The recurrence rate of hepatocellular carcinoma (HCC) is more than 70% at 5 years [8]. This is due to the cancer-specific characteristics which result in the cancer generally leaving the primary site very early, even when the tumor is ≤ 5 mm in diameter, and many remaining cancer cells are dormant and undetectable [9]. Several groups have reported the remaining tumor cells in the bloodstream even far after full treatments, likely 10 days postsurgical removal in gastrointestinal cancer patients [10] and 4 weeks after the end of chemotherapy in breast cancer patients [11].

On the other hand, in addition to ungovernable risk factors such as genetic susceptibility, environmental or psychological factors are considered crucial factors of cancer recurrence, as well as cancer initiation [12]. In particular, uncontrolled chronic psychological stress could induce unbalanced gene expression and cellular dysfunction, thus increasing the risk of recurrence- or metastasis-favorable tumor microenvironments such as macrophage infiltration, proangiogenesis, epithelial-mesenchymal transition, and tumor invasion [13,14]. Several studies have reported that psychological interventions for stress management improve clinical outcomes by modulating the neuroendocrine and immune systems to defend against tumor processes in breast cancer patients [15,16]. In addition, some programs for reducing psychological stress, including stress management and cognitive behavior therapy, are currently being performed at leading cancer centers [17].

However, there are conflicting claims about the contributing role of psychological stressors as an absolute risk factor for cancer recurrence [18,19]. Some case–control studies reported no relationship between psychological stress and the recurrence risk of cancer, including breast cancer and malignant melanoma [20,21]. In addition, one review reported that stressful life events are rather associated with decreased risk of colon and endometrial cancer incidence [22]. Various factors, including heterogeneity of different cancer type, individual sensitivity, and coping style to stress, could play a role in the causal association between stress and cancer recurrence risk, and should be investigated [23]. In fact, there are strong requirements to further provide clinicians and researchers with strong evidence regarding the effect of psychological stress on cancer recurrence and its proper management in cancer patients.

To determine the association between psychological stress and the risk of cancer recurrence, this review aims to systematically evaluate the relevant studies.

## 2. Materials and Methods

### 2.1. Search Strategy

The present review and meta-analysis were carried out according to the Preferred Reporting Items for Systematic Reviews and Meta-Analysis (PRISMA) guideline. The study was registered in PROSPERO; the registration number 284503. A systematic literature survey was conducted using three electronic databases: MEDLINE (PubMed), Embase, and Cochrane Central Register of Controlled Trials (CENTRAL). In addition, we manually searched the reference lists of relevant articles. We included all research articles published until 31 May 2021 and studies on humans without limitations, provided they were written in English. Both controlled terminology (MeSH and Emtree) and free text word searching were applied. A combination of search terms and key words included terms related to psychological stress (psychological stress, psychosocial stress, chronic stress, mental stress, distress, life events, job stress, work stress, death of family, loss of partner, depression, and anxiety) and outcomes (recurrence, relapse of neoplasm or cancer or tumor or carcinoma) and their combination. Two authors (H.-M.O. and C.-G.S.) systematically screened search results and independently reviewed each record in selection process.

### 2.2. Selection Criteria

We included studies researching the association of psychological stress with cancer recurrence. To be included, the impact of psychological stress or psychological state on the outcomes had to be analyzed by effect measures such as hazard ratios (HRs) or relative risks (RRs). We excluded nonclinical-based studies, studies on fear of cancer recurrence, and studies focusing on cancer-related biological aspects of psychological stress. Articles with no full text but not language limitations were excluded.

### 2.3. Data Extraction and Quality Assessment

The following details were extracted: name of first author, year of publication, country, type of cancer, number of participants, mean age, type of psychological stress, timing of stress measurement, follow-up period, type of intervention, type of stress measurement tool, number of cancer recurrences, HRs or RRs with associated 95% confidence intervals, and adjustments (Table 1). The assessment of cancer recurrence from each study based on medical records through conventional follow-up examination.

We assessed the quality of the observational studies by the Newcastle Ottawa Scale (NOS) [24]. For randomized controlled trials (RCTs), we assessed the methodological quality with the risk of bias tool (ROB 2.0) [25].

### 2.4. Review Process and Meta Analysis

We conducted a systematic review of these studies regarding the clinical associations between psychological stress and cancer recurrence by dividing them depending on the study design: cohort studies and RCTs. For the included cohort studies, meta-analysis could not be carried out due to the heterogeneity of each study, while meta-analysis was performed for RCTs using adjusted HRs with 95% confidence intervals, as mentioned in the original articles. By applying the generic inverse variance method, we calculated a pooled HR with 95% CI based on a random effects model according to the DerSimonian and Laird method [26]. Heterogeneity between the studies was investigated using Higgins I^2^, which measures the percentage of the total variation across studies that is due to heterogeneity [27]. RevMan version 5.0 (http://www.tech.cochrane.org/revman (accessed on 15 August 2021) was used for this analysis.

## 3. Results

### 3.1. Characteristics of the Included Studies

Among the 35 studies identified in the initial screening, 6 articles (4 cohort studies and 2 RCTs) met the inclusion criteria, as summarized in Figure 1. Detailed information on the included studies is provided in Table 1.

Five studies were for patients with breast cancer, and one study was for patients with HCC. The total number of patients participating was 26,329, including 79 males and 26,250 females. The six studies provided the mean age of the participating patients, and their total mean age was 53.5 ± 2.9. Two studies were published between 2000 and 2009, and four studies were published after 2010 from three countries, including the USA (*n* = 3), Denmark (*n* = 2), and China (*n* = 1). Regarding methodological quality, two studies were high-quality (NOS scores = 7), while the other two were relatively low-quality (NOS scores = 6). Two RCTs were judged to have ‘some concerns’ according to the RoB 2.0 tool (Appendix A).

### 3.2. Features of Psychological Stress and Interventions in Cohort and RCT Studies

In four cohort studies, seven observational analyses were conducted according to six different stress conditions: anxiety (in two studies), depression, loss of partner, emotional health, and mental health. Except for one study on anxiety in patients with HCC, all were observed in patients with breast cancer. Two RCTs applied ‘structured group meetings’ for 4 weeks or ‘cognitive-based stress management (CBSM)’ for 10 weeks as an intervention to patients with breast cancer. Except for one study (3 data points used RRs), the results were expressed as HRs (Table 1). If the HR/RR is 1.5 then the relative risk of recurrence in one group is 1.5 times higher than the risk of recurrence in the other group.

### 3.3. Psychological Stress and Risk of Tumor Recurrence from 4 Cohort Studies

Two (4 data points) of four cohort studies (8 data points) reported positive associations between psychologic/emotional distress and the risk of cancer recurrence [28,31], while others showed conflicting data (mental health vs. hostility) [29] or a reverse correlation (2 data points for loss of partner) [30]. Two data points for anxiety showed a significant increase in the HR 1.04 (*p* = 0.040) in patients with HCC [31] and RR 1.19 (*p* = 0.0281) in patients with breast cancer [28]. One cohort study in breast cancer patients reported a positive correlation between depression level and recurrence risk (RR 1.19, *p* = 0.1367) but a reverse correlation between good emotional heath and recurrence risk (RR 0.80, *p* = 0.0028) [28].

### 3.4. Psychological Stress and Risk of Tumor Recurrence from RCTs Using Adjusted Data

In two RCTs that applied structured group meetings or CBSM, structured group meetings significantly reduced the risk for cancer recurrence compared with the ‘assessment only’ group (HR = 0.55; 95% CI 0.32–0.96) [32], and CBSM also significantly reduced the cancer recurrence risk compared with ‘1-day psychoeducation’ (HR = 0.45; 95% CI 0.17–1.18) [33]. A meta-analysis proved the positive effects of these psychotherapies on the reduction in cancer recurrence risk in breast cancer patients (HR = 0.52; 95% CI 0.33–0.84, Figure 2).

## 4. Discussion

Psychological stress is very prevalent in up to 28% of the general population [34], and it has emerged as a significant health-related risk factor for various diseases, including cardiovascular disease, depression, and cancer [35]. As its underlying possible mechanisms, psychological stress in daily life is known to disrupt endocrine and autonomic homeostasis and lead to chronic inflammation [36]. In particular, cancer patients are more likely to suffer from psychological stress, and 31% of cancer patients show psychological stress-related clinical symptoms such as depression, anxiety, and psychotic disorders [37,38].

One previous study narratively reviewed a total of 15 studies (published from 1979 to 2012) investigating the association between psychological stress and cancer recurrence, which resulted in no relationship between these factors [39]. This review, however, only listed the articles without conducting the systematic analysis of statistical data. Despite the clinical importance of the effects of psychological stress on cancer recurrence, only limited evidence exists for a causal relationship between psychological stress and cancer recurrence [39]. As we showed here, only a total of six studies, to the best of our knowledge, have been performed to answer the question statistically. This may be due to methodological difficulties in designing and conducting studies, including ambiguity of defining and measuring stressed/unstressed levels, various types of psychological stress, individual personality coping with stress, and long-term requirements [40].

The present study extracted 10 statistics-based data points from the 4 cohorts and 2 RCTs for mainly breast cancer, except for 1 for HCC. We found that four cohorts and two RCTs presented significant effects of psychological stress on the risk for cancer recurrence. In the cohort studies, the risk for cancer recurrence was analyzed with four psychological stress-related scores (two ‘anxiety’ [28,31], one ‘depression’ [28], and one ‘hostility’ [29]), which showed a positive relationship. Two psychological health conditions, however, showed conflicting results; a higher level of ‘emotional health’ produced a lower risk of cancer recurrence, while ‘mental health’ showed the opposite [28,29]. In general, death of the spouse is known as one of the most stressful life events that triggers depressive symptoms and grief [41]; additionally, one cohort study showed contradictory results, with no increase in cancer recurrence risk, rather than a reduced likelihood in the group with ‘loss of partner’ both before and after diagnosis of breast cancer [30]. Indeed, there are some data for higher susceptibility in women against marital stress compared to job stress in men [42,43,44]. Psychosocial considerations for the conflicting impact of bereavement on the risk of cancer recurrence should be further studied.

Although the psychological stress generally leads to systemic responses to subjects’ psychological stress applied to all cancers, each cancer type has divergent biological characteristics depending on cancer type or histological subtype [45]. Accordingly, we hypothesized that the association between stress and the risk of cancer recurrence would be different depending on cancer type. Patients with breast cancer might be more sensitive to psychological stress, as women are prone to more daily/chronic stress, and there is evidence they may have a lower ability to cope with stress than men [46]. Of the included studies, five cohorts were all female with breast cancer, and one was HCC patients, more than half of which were male (79 males out of 110 patients). However, in our results, we did not find a significant difference in the risk of cancer recurrence between breast cancer and HCC. Further study to investigate stress-related risk on different types of cancer is needed.

Cohort studies have general limitations, such as a long study period, distortion by dropout, and difficulties in controlling the quantity and quality of the data [47]. In particular, stress-related cohort studies have many critical issues, including the heterogeneous types of stress and tools and time points of its measurements [35]. In contrast, an RCT-derived study would be able to answer the specific question [48]. A two-RCT-derived meta-analysis proved that stress-management therapies (CBSM and structured group meetings) had a positive effect on cancer recurrence risk in patients with postoperative breast cancer (HR = 0.52; 95% CI 0.33–0.84, Figure 2) [32,33]. CBSM teaches relaxation methods, including adjusting patients’ outlook, cognitive assessment, and coping strategies [49]. Previous clinical studies showed that CBSM led to positive psychosocial and physiological changes (lower cortisol and higher interleukin-2 and interferon-γ levels in serum) in stressed cancer patients [50,51]. Structured group meetings are one type of cognitive behavioral therapy to reduce negative emotional stress, including anxiety, depression, or fatigue, and have been applied to patients with various disorders/diseases, including cancer [52,53].

In addition to some examples from several hospitals, clinical management to relieve psychological stress is rarely implemented. As shown in our present systematic review, there are anticipating data supporting the necessity of stress management in cancer patients, not only to help QoL, but also to reduce cancer recurrence. Five of six studies were conducted for breast cancer, which has relatively low mortality and is easy to observe over a long-term period [54]. Further studies are still required based on well-designed RCTs for diverse cancer types, especially by measuring comprehensive stress levels through a multidisciplinary approach.

There are several limitations in our present study. First, we included a small number of studies, with four cohort studies and two RCTs. Our study has a possibility of publication bias. Second, only two types of cancer were included in this review, which makes our results difficult to be generalized for diverse cancer types. Lastly, measurement of stress is inconsistent in each study due to the different types of stress and grouping stress level. However, our study is the first to comprehensively produce the association between psychological stress and the risk of cancer recurrence. This systematic review would be valuable to provide reference data to clinicians and scientists for further studies.

## 5. Conclusions

Our systematic review suggests that psychological stress is moderately linked to the risk of cancer recurrence. Moreover, two RCTs strongly support the beneficial effects of stress-management therapies on reducing cancer recurrence in patients, especially after resection of breast cancer. Despite the limitations of the relatively small amount of data, our results can serve as a basis for future studies.

## Figures and Tables

**Figure 1 cancers-13-05816-f001:**
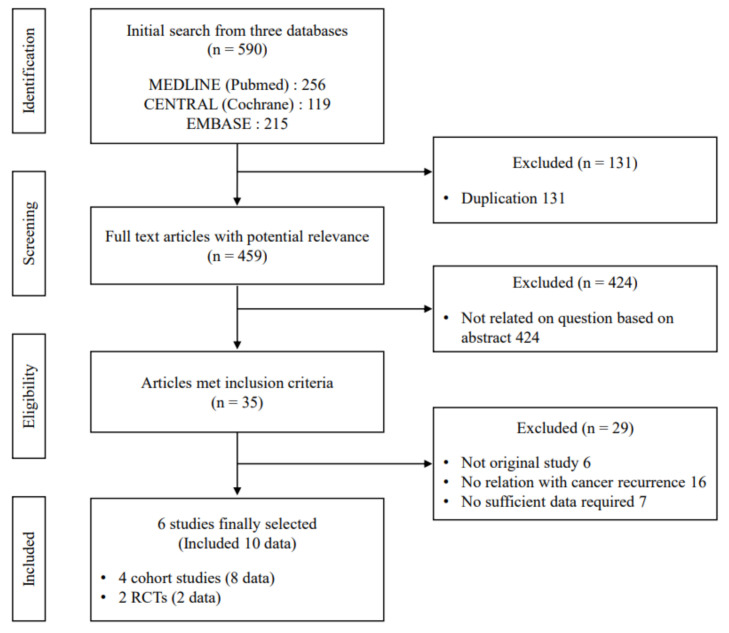
Flow diagram of literature search process.

**Figure 2 cancers-13-05816-f002:**
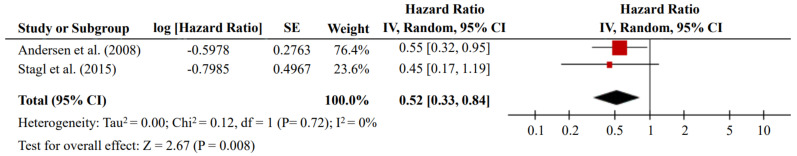
Meta−analysis of 2 RCTs. The size of the rectangle increases as the number of samples in each study increases and the confidence interval narrows. Diamond represents the overall outcome of the meta−analysis as a weighted average of included studies.

**Table 1 cancers-13-05816-t001:** Summary of 6 studies included.

Author (Year), Country	Cancer Type N. of Participants Mean Age (SD)	Stress Measurement Tool	Psychological Stress /Intervention	Timing of Stress Measurement	Median Observation/Follow-Up Period (Years)	Stress High vs. Low /Intervention vs. Control	N. of Recurrence	Adjusted HR or RR (95% CI)	Adjustments
**Cohort studies**									
Groenvold et al. (2007), Denmark [28]	Breast cancer 1,588 52.4 ± 10.3	EORTC QLQ-C30 HADS HADS	Emotional health Anxiety Depression	7 weeks after surgery	12.9	EORTC QLQ-C30 score: 83–100 vs. 0–75 HADS score: 8–21 vs. 0–7 HADS score: 8–21 vs. 0–7	761 (No information on each group)	RR = 0.80 (0.69–0.93) RR = 1.19 (1.02–1.39) RR = 1.19 (0.95–1.50)	age, adjuvant treatment regimen, grading of anaplasia, histopathological diagnosis, local radiotherapy, menopausal status, number of tumor positive axillary nodes, social class type of operation, tumor receptor status, tumor size
Saquib et al. (2011), USA [29]	Breast cancer 2,967 53.3 ± 8.9	SF-36 CMHS	Mental health Hostility	At first clinic visit	7.3	SF-36 score: 90.6–100 vs. 0–63.3 CMHS score: 6–13 vs. 0	492 (No information on each group)	HR = 1.21 (0.85–1.72) HR = 1.24 (0.92–1.68)	age, anti-estrogen use, hot flashes, menopausal status, physical activity, race/ethnicity, clinical site, time between cancer diagnosis and study entry, randomization group, tumor type, tumor grade, tumor stage
Olsen et al. (2012), Denmark [30]	Breast cancer 21,213 (Alive 19,312 vs. Before 762, After 1,139) 59.3 ± 4.11	-	Loss of partner	-	7.7	Loss of partner vs. partner alive	2,779 (Alive 2,635 vs. Before diagnosis 59 After diagnosis 85)	HR = 0.82 (0.61–1.09) HR = 0.98 (0.73–1.18)	age, disposable income, comorbidity, highest-attained educational level, hormone receptor status, number of tumor-positive lymph-nodes, period of diagnosis, tumor size, tumor grade
Liu et al. (2016), China [31]	HCC 110 (High 62 vs. Low 48) 54.5, range 36-76	HAMA	Anxiety	3 months after surgery	4.0	HAMA score: 17–56 vs 0–16	44 (High 31 vs. Low 13)	HR = 1.04 (1.00–1.10)	No information
**RCTs**									
Andersen et al. (2008), USA [32]	Breast cancer 211 (Inter. 103 vs. Cont. 109) 55.5 ± 11.9/52.2 ± 10.8		Structured group meeting (48 weeks)		11	48 weeks of structured group meeting vs. Assessment only	62 (Inter. 29 vs. Cont. 33) *	HR = 0.55 (0.32–0.96)	age, chemotherapy, hormonal therapies, lymph node status, hormone receptor status, histologic grade, histologic type, Karnofsky performance status, menopausal status, POMS score, radiotherapy, surgery type, tumor size
Stagl et al. (2015), USA [33]	Breast cancer 240 (Inter. 120 vs. Cont. 120) 49.7 ± 9.0/51.0 ± 9.1		CBSM (10 weeks)		11	10 weeks of CBSM vs. 1-day psychoeducation	47 (Inter. 24 vs. Cont. 23) *	HR = 0.45 (0.17–1.18)	age, disease stage, Her2/neu status, hormonal therapies, tumor size

HR; hazard ratio, RR; relative risk, CI; confidence interval, EORTC QLQ-C30; European organization for research and treatment of cancer-quality of life, HADS; hospital anxiety and depression scale, HAMA; Hamilton rating scale for anxiety, SF-36; short form 36-item health survey, CMHS; Cook-Medley hostility scale, RCT; randomized controlled trial, POMS; profile of mood state, CBSM; cognitive based stress management. * Although there was no significant difference in the number of recurrences, patients with intervention have a lower risk of recurrence due to delayed time to recurrence.

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
