# Peer review of "The Risk of Psychological Stress on Cancer Recurrence: A Systematic Review"

_cancers, 2021, doi:10.3390/cancers13225816_

Round 1

Reviewer 1 Report

  1. Type of review: The title states „systematic review“ but line 101 describes it as „narrative review“. This need clarification. If the article claims to be a “systematic review” then it is necessary to demonstrate that all formal requirements of a “systematic review” are fulfilled.
  2. Both, introduction and discussion could benefit from some more details regarding background and the controversial discussion about stress as a risk factor for carcinogenesis as well as tumor progression/recurrence.
  3. Search strategy (line 76): The authors state that articles published until May 2021 were included. It remains unclear whether all studies prior May 2021 would potentially be eligible or whether eligible articles had to be published after a certain date.
  4. Data extraction (line 91): It appears that no information regarding potential confounding factors as well as length of follow-up period has been extracted. Failure to take into account this information will result in insufficient quality assessment.
  5. Statistical analysis (line 100)/Table 1: No information is given which cut-points were applied to define stress as a dichotomous independent variable in studies using ordinal or likert scales. It is unclear to the reader what the adjusted HR/RR mean. How is the “change in estimate” defined? How do you define “exposed” and “control” in the various studies? For which confounders were the effect measures “adjusted” for in each study?
  6. Methods and Table 1: How was recurrence assessed in each study? How long was the follow-up period? What about the potential for selective survival and immortal time bias?
  7. What is the evidence for a publication bias? I am surprised to learn that only a small number of articles shall be published.
  8. Line 136: Avoid using terminology claiming causality such as “effect”.
  9. Line 193/194: RCTs are able to answer the question whether an intervention (e.g. stress reduction) might have some effect but not whether the addressed factor (here: stress) is a true causal factor. Please revise.

Author Response

First of all, I would like to thank reviewer for the thorough review and helpful comments for our manuscript (ID: cancers-1449172) submitted previously. We have carefully read the comments and revised thoroughly our manuscript titled “The Risk of Psychological Stress on Cancer Recurrence: A Systematic Review”. All changes had been highlighted and we added the responses to reviewers’ comments. We look forward to your positive response.

Reviewer’s comment

  1. Type of review: The title states “systematic review” but line 101 describes it as “narrative review”. This need clarification. If the article claims to be a “systematic review” then it is necessary to demonstrate that all formal requirements of a “systematic review” are fulfilled.

=> We really appreciate reviewer for the detail and professional comments. The word “narrative review” was a typo. We have conducted systematic review according to the PRISMA after PROSPERO registration of systematic review (Registration number : 284503). We revised our current manuscript.

  1. Both, introduction and discussion could benefit from some more details regarding background and the controversial discussion about stress as a risk factor for carcinogenesis as well as tumor progression/recurrence.

=> As reviewer suggested, we have reinforced the description on stress as a risk factor for carcinogenesis, cancer progression/recurrence and its controversies.

  1. Search strategy (line 76): The authors state that articles published until May 2021 were included. It remains unclear whether all studies prior May 2021 would potentially be eligible or whether eligible articles had to be published after a certain date.

=> We searched all studies published by 31th May 2021 without limitation of the earliest year. We have improved the sentence according to reviewer’s comment in current revised manuscript.

  1. Data extraction (line 91): It appears that no information regarding potential confounding factors as well as length of follow-up period has been extracted. Failure to take into account this information will result in insufficient quality assessment.

=> We fully agree with reviewer’s opinion. As reviewer indicated, we added the information of adjustments and length of follow-up period for all studies in Table 1.

  1. Statistical analysis (line 100)/Table 1: No information is given which cut-points were applied to define stress as a dichotomous independent variable in studies using ordinal or likert scales. It is unclear to the reader what the adjusted HR/RR mean. How is the “change in estimate” defined? How do you define “exposed” and “control” in the various studies? For which confounders were the effect measures “adjusted” for in each study?

=> We sincerely appreciate reviewer for professional comments. As reviewer indicated, we have described the cut-points that distinguish the stress high vs low group (“exposed” vs. “control”) in Table 1.

=> In our study, high HR/RR means higher risk of recurrence. As reviewer suggested, we have added the information about meaning of HR/RR in revised manuscript.

=> We have added the confounding factor, but how the change in estimates was defined in each study was not explained. We also have added the information of adjustments in Table 1.

  1. Methods and Table 1: How was recurrence assessed in each study? How long was the follow-up period? What about the potential for selective survival and immortal time bias?

=> We appreciate reviewer for the helpful comments. Assessment of recurrence in all included studies except study of Liu et al. (2016) were based on information obtained from medical records. Liu et al. (2016) diagnosed intrahepatic recurrence with two factors: 1) the histopathological detection of tumor tissue in patients undergoing repeated hepatic resection and 2) the characteristic appearance of HCC in US, CT, MRI, and hepatic angiography scans. All patients were screened with US every 3 months, with CT or MRI every 6 months, and with hepatic angiography when recurrence was suspected.

=> As reviewer indicated above, we have described the follow-up period of each studies in revised Figure 1.

=> The studies selected to our review included the patients after surgical removal was completed. The main causes of death in cancer patients are metastasis and recurrence as we described in ‘Introduction’, so the potential of selective survival or immortal time bias is low.

  1. What is the evidence for a publication bias? I am surprised to learn that only a small number of articles shall be published.

=> As reviewer indicated, we are not confident of no publication bias. We are also surprised about the small number of articles. We however thought that it was due to the difficulties in conducting studies for stress-related risk of tumor recurrence, including ambiguity of defining and measuring stressed/unstressed levels, various types of psychological stress, individual personality coping with stress and long-term requirements. We have discussed the possibility of publication bias in this revised manuscript. 

  1. Line 136: Avoid using terminology claiming causality such as “effect”.

=> We appreciate reviewer for professional comment. As reviewer indicated, we have corrected the controversial terminology in our revised manuscript.

  1. Line 193/194: RCTs are able to answer the question whether an intervention (e.g. stress reduction) might have some effect but not whether the addressed factor (here: stress) is a true causal factor. Please revise.

=> Once again, we sincerely appreciate reviewer for the professional correction. We have revised the sentences.

Reviewer 2 Report

I congratulate the authors for approaching the very interesting and clinically relevant topic of the risk of psychological stress on the cancer recurrence.

Major comments

General comments

- This review is based on a small number of studies (4 cohort studies and 2 RCTs), which limits the relevance of the results and conclusions obtained.

- The authors consider in the review five studies related to breast cancer and 1 study addressing HCC. This could be a bias in the results obtained because breast cancer and liver cancer are two different clinical entities in terms of therapy, patient characteristics, recurrence rate and cure rate. The authors should clarify this in the review.

Introduction

- Given that the review includes five studies targeting breast cancer patients and one study involving HCC patients, the authors should mention the extent to which the type of cancer influences patients' psychological stress levels.

- The introduction should provide information on the recurrence rate of HCC (the authors mention the recurrence rates of breast, colorectal and gastrointestinal cancer, but not the recurrence rate of HCC, although this type of cancer is considered in the review).

Material and methods

- Search strategy - the authors must state the time period for which the literature search was made. In the article the authors mention only "studies published until May 2021" (L76).

- The authors mention that “studies without limitations in the English language” have been taken into account. I understand from this statement that articles written in languages other than English have been included in the review. In this case, the authors should specify in which languages the articles included in the review were written. How was the translation done? Who and how verified the correctness of the translation?

- How many reviewers screened each record? Did the reviewers work independently? How did they resolve the disagreements among them, if any?

Results

- The authors need to review Fig. 1- Flow diagram of literature search process. In the flow diagram it is mentioned that after screening (after excluding duplicates) 459 articles remained; of these, 431 were excluded for the reason “not related on questions based on abstract”, which means that 28 articles remained and not 35 as mentioned in the text and in the flow diagram.

- Authors should explain the relevance of assessing the methodological quality of the studies included in the review for the results of the review (L124-127)

Discussions

- The authors must discuss the reason(s) for which the results obtained by them are totally different from those reported in the review of 15 articles published in the period 1979-2012.

Minor comments

  • The authors must specify the type of review they have carried out, as the text initially states that a narrative review has been carried out (L101), and subsequently states that a systematic review has been carried out (L 206); the title also mentions the term "systematic review". Therefore, clarification is needed on this issue.
  • At L129-130 it is written that: [...] 6 different stress conditions: anxiety, depression, loss of partner, emotional health, and mental health. Although the authors refer to 6 different stress conditions, only 5 are listed in the text.
  • L 143-144 - the authors specify the existence of a "reverse correlation between good emotional health", but they do not specify the second term of the correlation.

Author Response

First of all, I would like to thank reviewer for the thorough review and helpful comments for our manuscript (ID: cancers-1449172) submitted previously. We have carefully read the comments and revised thoroughly our manuscript titled “The Risk of Psychological Stress on Cancer Recurrence: A Systematic Review”. All changes had been highlighted and we added the responses to reviewers’ comments. We look forward to your positive response.

Reviewer’s comment

I congratulate the authors for approaching the very interesting and clinically relevant topic of the risk of psychological stress on the cancer recurrence.

=> We sincerely appreciate reviewer for the thorough review.

General comments

This review is based on a small number of studies (4 cohort studies and 2 RCTs), which limits the relevance of the results and conclusions obtained.

The authors consider in the review five studies related to breast cancer and 1 study addressing HCC. This could be a bias in the results obtained because breast cancer and liver cancer are two different clinical entities in terms of therapy, patient characteristics, recurrence rate and cure rate. The authors should clarify this in the review.

=> We fully understand the reviewer’s critique, and really appreciate reviewer for the professional comments. As reviewer indicated, our study has limitations, especially due to the small number of eligible data as well as two heterogenous characteristics (breast cancer vs. HCC patients). We have discussed further detail about this issue in revised manuscript.

Given that the review includes five studies targeting breast cancer patients and one study involving HCC patients, the authors should mention the extent to which the type of cancer influences patients' psychological stress levels.

=> We appreciate reviewer for professional comment. There are controversies that the influence of psychological stress on cancer differs depending on the type of cancer. We have discussed this in our revised manuscript.

The introduction should provide information on the recurrence rate of HCC

=> We appreciate reviewer for detailed comment. As reviewer indicated, our current data have a limitation. We have added the statistical data statements of recurrence rate of HCC as reviewer indicated.

The authors must state the time period for which the literature search was made. In the article the authors mention only "studies published until May 2021" (L76).

=> Literature search had been conducted from April to May 2021 in our study. We included all studies meeting our inclusion/exclusion criteria published before May 2021. We have improved the sentence according to reviewer’s comment in current revised manuscript.

The authors mention that “studies without limitations in the English language” have been taken into account. I understand from this statement that articles written in languages other than English have been included in the review. In this case, the authors should specify in which languages the articles included in the review were written. How was the translation done? Who and how verified the correctness of the translation?

=> We apologize for misrepresentation of selection and inclusion process. All languages ​​were considered in the selection process, but all studies in our review were conducted and written in English only.

How many reviewers screened each record? Did the reviewers work independently? How did they resolve the disagreements among them, if any?

=> Two authors (H.-M.O and C.-G.S) systematically screened search results and independently reviewed each record in selection process. Any discordances regarding study inclusion between these two authors were settled after much discussion. We have added the sentences in revised manuscript.

The authors need to review Fig. 1- Flow diagram of literature search process. In the flow diagram it is mentioned that after screening (after excluding duplicates) 459 articles remained; of these, 431 were excluded for the reason “not related on questions based on abstract”, which means that 28 articles remained and not 35 as mentioned in the text and in the flow diagram

=> Thank reviewer for the helpful advice. We have corrected them in Figure 1.

Authors should explain the relevance of assessing the methodological quality of the studies included in the review for the results of the review (L124-127)

=> We really thank reviewer for the professional comment. There was no difference in the results according to the whether the study was of low or high quality.

The authors must discuss the reason(s) for which the results obtained by them are totally different from those reported in the review of 15 articles published in the period 1979-2012.

=> We really appreciate reviewer for the professional comment. In our study, we included only 6 papers with statistical data through quality evaluation of each study and the previous study included all papers that described only small relevance regardless of the study quality. We could not establish a clear causal relationship, but provide summarized information using statistical data.

The authors must specify the type of review they have carried out, as the text initially states that a narrative review has been carried out (L101), and subsequently states that a systematic review has been carried out (L 206); the title also mentions the term "systematic review". Therefore, clarification is needed on this issue.

=> We really appreciate reviewer for the helpful comments. The word “narrative review” was a typo. We have conducted systematic review according to the PRISMA after PROSPERO registration of systematic review (Registration number: 284503). We revised our current manuscript.

At L129-130 it is written that: [...] 6 different stress conditions: anxiety, depression, loss of partner, emotional health, and mental health. Although the authors refer to 6 different stress conditions, only 5 are listed in the text.

=> There were 2 data that studied anxiety in different studies, so only 5 were listed in the text. We have revised our current manuscript.

L 143-144 - the authors specify the existence of a "reverse correlation between good emotional health", but they do not specify the second term of the correlation.

=> We have corrected them.

Round 2

Reviewer 1 Report

Thank you very much for the careful revision. My comments have been adequately addressed except for my concerns regarding "causal terminology": Replacing "effects" by "influences" is not solving the problem of claiming causality when you merely observe associations. Please revise.

Minor typo in line 142: "(in 2 study)" should read "(in 2 studies)" .

Author Response

Thank you very much for the careful revision. My comments have been adequately addressed except for my concerns regarding "causal terminology": Replacing "effects" by "influences" is not solving the problem of claiming causality when you merely observe associations. Please revise.

=> We really appreciate reviewer for the professional review. Thanks to the thorough review, we have improved the quality of our paper. As reviewer indicated, we have deleted the word “influences” and described only the associations without claiming causality.

Minor typo in line 142: "(in 2 study)" should read "(in 2 studies)".

=> Once again, we sincerely appreciate reviewer for the detailed correction. As reviewer indicated, we have corrected the typo in our current manuscript.

Reviewer 2 Report

I thank the authors for carefully reviewing their article, taking into account all the comments I made in my first report.

However, I suggest the authors revise the text to correct the typos. E.g.:  Line 142- (in 2 study); Line 212- 110 patient; Line 242- caner.

Author Response

I thank the authors for carefully reviewing their article, taking into account all the comments I made in my first report.

=> We really appreciate reviewer for the professional review. Thanks to the thorough review, we have improved the quality of our paper.

However, I suggest the authors revise the text to correct the typos. E.g.:  Line 142- (in 2 study); Line 212- 110 patient; Line 242- caner.

=> Once again, we sincerely appreciate reviewer for the detailed correction. As reviewer indicated, we have corrected the typo in our current manuscript.